# Mediastinal Gray-Zone Lymphoma: Still an Open Issue

**Stefano Pileri** [1,2,*], **Valentina Tabanelli** [1], **Roberto Chiarle** [1,3], **Angelica Calleri** [1], **Federica Melle** [1], **Giovanna Motta** [1], **Maria Rosaria Sapienza** [1], **Elena Sabattini** [4], **Pier Luigi Zinzani** [2,5] and **Enrico Derenzini** [6,7]

1   Division of Haematopathology, European Institute of Oncology IRCCS, Via Ripamonti 435, 20141 Milan, Italy; valentina.tabanelli@ieo.it (V.T.); roberto.chiarle@ieo.it (R.C.); angelica.calleri@ieo.it (A.C.); federica.melle@ieo.it (F.M.); giovanna.motta@ieo.it (G.M.); mariarosaria.sapienza@ieo.it (M.R.S.)
2   Department of Medical and Surgical Sciences, Bologna University, Via Massarenti 9, 40138 Bologna, Italy; pierluigi.zinzani@unibo.it
3   Department of Biomedical Sciences and Human Oncology, Turin University, Via Santena 7, 10126 Turin, Italy
4   Unit of Haematopathology, University-Hospital S. Orsola Polyclinic IRCCS, Via Massarenti 9, 40138 Bologna, Italy; elena.sabattini@aosp.bo.it
5   Institute of Haematology "L. and A. Seràgnoli", University-Hospital S. Orsola Polyclinic IRCCS, Via Massarenti 9, 40138 Bologna, Italy
6   Division of Haemato-Oncology, European Institute of Oncology IRCCS, Via Ripamonti 435, 20141 Milan, Italy; enrico.derenzini@ieo.it
7   Department of Health Sciences, University of Milan, Via di Rudinì 8, 20146 Milan, Italy
*   Correspondence: stefano.pileri@unibo.it or stefano.pileri@ieo.it; Tel.: +39-02-57489251 or +39-348-2237355

**Abstract:** The concept of gray-zone lymphoma (GZL) has been progressively refined since its introduction in the literature in 1998. For several years, it was applied to a rather broad spectrum of conditions, posing the problem of the differential diagnosis between any type of Hodgkin lymphoma (HL) and diffuse large B-cell lymphoma, with special reference to primary mediastinal forms (PMBL). Officially recognised as a provisional entity in the 4th and revised 4th editions of the WHO Classification of Tumour of Haematopoietic and Lymphoid Tissues with the term "B-cell lymphoma unclassifiable with features intermediate between diffuse large B-cell lymphoma and classic Hodgkin lymphoma", it was limited to tumours showing either morphologic features reminiscent of classic HL (CHL) but carrying a complete B-cell phenotype or conversely provided with a PMBL morphology yet revealing CHL phenotypic characteristics. The definition of GZL has been further revised in the recently published International Lymphoma Classification and 5th edition of the WHO Classification of Haematolymphoid Tumours, which have limited it to mediastinal neoplasms (MGZL) based on emerging molecular evidence. The aim of this review is to critically discuss the issue of MGZL, as well as in light of the suboptimal response to current therapies.

**Keywords:** mediastinal gray-zone lymphoma; classic Hodgkin lymphoma; primary mediastinal B-cell lymphoma; diffuse large B-cell lymphoma; NOS; EBV-positive diffuse large B-cell lymphoma; NOS; morphology; phenotype; gene expression profile; mutational landscape

## 1. Introduction

Mediastinal gray-zone lymphoma (MGZL) represents a problem for both pathologists and clinicians because of the rarity of the condition, the rather subjective diagnostic criteria, the lack of standard therapy, and its aggressive clinical course. The aim of the present review is to discuss (a) the evolution of the concept of MGZL; (b) potential limitations of tissue sampling; (c) the more restrictive diagnostic criteria introduced by the most recent classification proposals; (d) the novel pathobiological information provided by high-throughput technologies, and (e) therapeutic perspectives.

## 2. Evolution of the Concept of Gray-Zone Lymphoma

The term gray-zone lymphoma (GZL), originally proposed by Rüdiger et al. in 1998, referred to a "distinct subset of cases representing mediastinal large B-cell lymphomas

with features of Hodgkin's lymphoma" [1]. In 2005, Poppema et al. underlined that "classic Hodgkin lymphoma (CHL) and at least a proportion of primary mediastinal B-cell lymphoma (PMBL) are derived from B cells at similar stages of differentiation and share common pathogenetic features" [2]. Such a statement was supported by the evidence published two years previously by Rosenwald et al., showing that the gene expression profile of PMBL was similar to the profile of CHL, with the exception of the preservation of the B-cell programme, repressed in CHL [3]. Through time, the concept of GZL was further expanded to tumours showing borderline characteristics between HL and large B-cell lymphoma (LBCL) by also including lymphocyte-rich CHL, nodular lymphocyte predominant Hodgkin lymphoma (NLPHL), and T-cell/histiocyte-rich LBCL (THRLBCL), as well as cases of ambiguous morphology and phenotype located at extra-mediastinal sites and/or EBV positive [4].

In 2008 and 2017, the fourth and revised fourth editions of the WHO Classification of Tumours of the Haematopoietic and Lymphoid Tissues introduced the provisional entity "B-cell lymphoma unclassifiable with features intermediate between diffuse large B-cell lymphoma and classic Hodgkin lymphoma", which included two subtypes, characterised by morphologic features reminiscent of CHL but carrying a complete B-cell phenotype or conversely provided with a PMBL morphology but revealing a CHL phenotypic profile (with strong CD30 and CD15 positivity, together with a lack of most if not all B-cell markers as well as of CD45) [5,6].

Over the last few years, several important contributions have been produced thanks to the extensive application of high-throughput molecular techniques as well as to the attempt to better understand the morphologic and phenotypic spectrum of primarily mediastinal GZLs (MGZLs) [7–9].

Currently, both the International Consensus Classification (ICC) and the 5th Edition of the WHO Classification (WHO-HEMA5) agree in limiting the concept of GZL to a mediastinal B-cell lymphoma with overlapping features between PMBL and CHL, especially nodular sclerosis CHL (NSCHL) [10,11]. Current evidence indicates that cases with morphologic and immunophenotypic features similar to MGZL but occurring outside and without the involvement of the mediastinum harbour different gene expression profiles and DNA alterations [12]. Hence, these cases are better classified as diffuse large B-cell lymphoma, not otherwise specified (DLBCL, NOS). Last but not least, cases with composite or sequential CHL and PMBL, which are rare but do exist, are not currently included in the setting of MGZL, since they need further studies to understand their natural history, as they may represent examples of tumours with high biological plasticity [2,13,14].

Concerning the possible relationships between MGZL and EBV infection, while the WHO-HEMA5 does not provide any firm indication, the ICC states that "nearly all patients with EBV-positive DLBCL, while they may harbour admixed Hodgkin/Reed-Sternberg-like cells, differ at the genomic level from patients with MGZL and should be retained within the category of EBV-positive DLBCL" [10,11].

Based on the more restrictive criteria introduced by the ICC and WHO-HEMA5, the present review will focus on MGZL.

### 3. Clinical Characteristics

According to Kritharis et al. and Pilishowska et al., the median age of patients with MGZL varies between 33 and 35 years, approximately two-thirds of whom are males [15,16]. Of these patients, nearly half have a bulky mediastinal mass, while a minority have extranodal involvement and only 13% stage IV disease. The mediastinal mass can cause superior vena cava syndrome or respiratory distress. Supraclavicular lymph nodes may be involved. These may be spread to the lung by direct extension, as well as to the liver, spleen, and bone marrow, in particular in relapsing or resistant (R/R) cases [15]. A different median age (46–48 years) and sex ratio (M/F = 1/1) were reported by Sarkozy et al. [8,9]. However, the latter authors provided a survey of their patients irrespective of whether they carried a mediastinal ("thymic") or a nonmediastinal ("nonthymic) tumour, an element that

may have influenced the reported clinical findings. In fact, when separating mediastinal from nonmediastinal cases, Pilichowska et al. found significant differences in terms of median age (35 vs. 51 years), stage I/II (89% vs. 46%), and bulky disease (44% vs. 0%) [16].

## 4. Biopsy Issues

The complexity of the diagnostic approach to MGZL is further hampered by the amount of bioptic material available. Mediastinal masses, which fulfil the criteria of bulky disease, are often sampled by needle biopsies. Compatibly with the condition of the patient, a wedge biopsy in the course of a mini-thoracotomy would be much more useful to solve the diagnostic problems. The latter comprise (1) the distinction of MGZL from PMBL, DLBCL, NOS, and CHL (including the syncytial variant as discussed below); (2) the application of the classification into four groups proposed by Sarkozy and coworkers reviewed in the following [7]; (3) the exclusion of a composite tumour, which the present definition of MGZL does not apply to; and (4) sampling problems, for instance, related to an exuberant sclerotic component [10,11]. The correct diagnosis represents a key point, because the therapies used for PMBL, CHL, or MGZL are different as are the results they produce. In particular, MGZL have shown the worst outcome even when treated with intensive regimens such as DA-EPOCH-R (see below) [17]. Finally, a fine needle aspirate cytology is totally inadequate for the diagnosis of MGZL [18]. Such an approach may provide useful information to exclude granulomatous disorders (e.g., sarcoidosis and tuberculosis), metastatic diseases, and T-lymphoblastic lymphoma.

## 5. Morphology and Phenotype

In spite of the criteria given by lymphoma classifications (WHO 4th, revised 4th, 5th, and ICC), the diagnosis of MGZL remains rather subjective, a fact that can lead to use of the term as a convenient basket category [5,6,10,11]. In a still ongoing study promoted by the Italian Lymphoma Foundation (FIL), most diagnoses of MGZL rendered at peripheral Italian Hospitals were not confirmed upon central review by an expert panel (unpublished data). This finding is in keeping with the reports of Pilichowska et al. and Sarkozy et al., who also found that only a proportion of GZLs (38% and 65%, respectively) submitted for central review were confirmed by a panel of experts [8,16].

Attempts have been made to define the different morphologic and phenotypic conditions to which the term MGZL can be applied.

In the *American Journal of Surgical Pathology*, Sarkozy et al. described four different groups (which they graded from zero to three) [7]. Group 0 showed a Hodgkin-like morphology with a scleronodular architecture and Reed–Sternberg cells (RSCs) in an inflammatory background. Tumoral cells expressed CD30 and showed strong and diffuse positivity for CD20 in 100% of them along with other B-cell markers, such as CD79a, PAX5/BSAP, and OCT2. Based on what was observed in the abovementioned FIL study, in Group 0, the CD15 staining was negative or occasionally weakly positive in a few neoplastic cells. In comparison with the previous one, Group 1 reveals a Hodgkin-like morphology with a more intermediate architecture, less fibrosis, and numerous sheets of large mononucleated cells in an inflammatory background. In addition to positivity for CD20 and other B-cell markers, such as BOB1 and OCT2, there was partial expression of CD15. Group 2 had a large B-cell morphology with some RSCs in an inflammatory background. Tumoral cells strongly expressed CD30 and CD15. PAX5 was moderately positive, while CD20 was negative. Other B-cell markers were variably expressed. Finally, Group 3 had a large B-cell morphology with fibrosis producing compartmentalisation, as observed in PMBL. Phenotypically, neoplastic cells strongly expressed CD20, CD30, and CD15. The stains for CD23 and MAL did not influence the subdivision into the four groups. The majority of the cases expressed PD-L1/PD-L2 and carried *CIITA* and *CD274* loci abnormalities on FISH analysis. One of the major limitations of the classification into four groups proposed by Sarkozy and coworkers was the inclusion of nonmediastinal cases, as well as of EBV-positive tumours, which are nowadays taken separate from MGZLs,

as mentioned above [7,10,11]. Three further comments should be made concerning the morpho–phenotypic pictures described by Sarkozy et al. [7]. First, the distinction among the four groups may be subjective. The ongoing FIL study has shown so far that the subclassification is not always easy to apply, also because of the variability of morphologic and phenotypic findings from field to field within the same case. In addition, one should bear in mind that CHL, PMBL, and MGZL can represent a continuum of diseases, which share a series of phenotypic and molecular characteristics and whose borders are arbitrarily drawn to some extent [8,9]. Secondly, Group 1–2's cases need to be distinguished from the syncytial variant of CHL that is characterised by solid sheets of neoplastic cells with a DLBCL-like morphology [5,6]. The diagnosis of such variant should be rendered only when tumoral elements show the typical phenotype of CHL [5,6,10,11]. Thirdly, the expression of PD-L1/PD-L2 is definitely stronger in MGZL than in nonmediastinal tumours in the past included in the broader definition of GZLs, the opposite being true for MHC1 [8].

The view of Sarkozy et al. is not shared by Egan and Pittaluga, who underlined that MGZL is characterised by varying degrees of discordance between morphology and immunophenotype, making its classification difficult [19]. Tumour cells are usually abundant, and although traditionally categorised as "PMBL-like" or "CHL-like", the cytomorphological spectrum is in between that seen in PMBL or CHL. Neoplastic cells may have a centroblastic or immunoblastic appearance but can be larger and more pleomorphic. Lacunar cells or RS-like cells can be detected in variable numbers and turn out to be more intermediate in size. The inflammatory milieu of CHL-like is often less apparent, with fewer eosinophils and plasma cells. Necrosis is variable but may be scarce or absent. The tumour architecture is usually diffuse, but occasionally a nodular growth pattern and coarse fibrosis may be found. The expression of PAX-5/BSAP, BOB.1, and OCT-2 is variable among MGZLs, but usually there is positivity for at least one B-cell marker in a panel of CD20, CD79a, and PAX5. The cases rich in centroblastic-appearing cells may lack or only partially express B-cell markers (such as CD20) but show positivity for CD30 or moderate–strong staining for CD15. Tumours with an abundance of RS-like cells may have a strong expression of multiple B-cell markers/transcription factors and CD45 and lack CD15 positivity. CD30 is usually expressed, although it may be weak and positivity for CD23 may also occur. MAL may also be positive irrespective of histology. EBER positivity has occasionally been reported, although it would exclude the diagnosis of MGZL according to the more restrictive criteria of the ICC, as cited above [10] (Figures 1 and 2).

The tumour microenvironment (TME) of MGZL would merit more in-depth studies, taking into consideration, however, that it can significantly vary based on the morphologic and molecular findings of Sarkozy and co-workers [7,8]. For instance, it would be useful to assess whether there is any predictive test that can herald therapeutic responses, as recently proposed for T-cell/histiocyte-rich large B-cell lymphoma (THRLBCL) [20]. The latter reference may appear out of context. However, a paper on gene expression profiling by Sarkozy et al. (see also below) focuses on the differential expression of several molecules in MGZL compared to NSCHL and PMBL (e.g., LAG3, PD1, and PD-L1), which also turns out to be deregulated in THRLBCL in comparison with DLBCL and NOS [8,20]. Thus, more attention to the TME of the different groups of MGZL, according to Sarkozy et al., might also lead to the discovery of novel therapeutic targets, especially by applying the tools for single cell analysis.

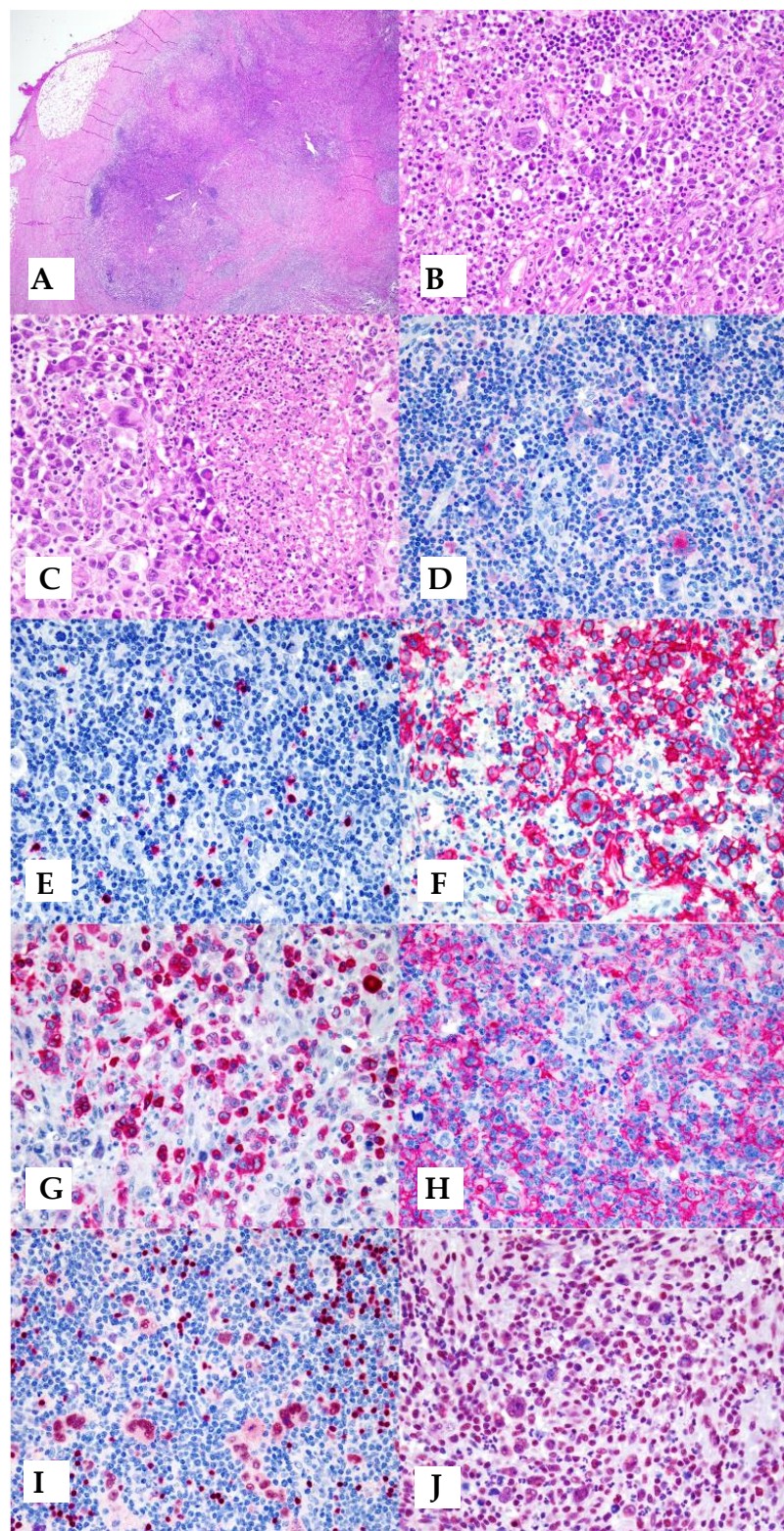

**Figure 1.** Example of MGZL with CHL-like morphology: (**A**) fibrosis with nodular growth, reminiscent of nodular sclerosis CHL; (**B**) Reed–Sternberg-like cells, as well as mononuclear neoplastic cells, comprised within a rich inflammatory milieu; (**C**) area with a sheet of neoplastic cells centred by necrosis; (**D**) CD30 is partly and variably expressed by neoplastic cells; (**E**) CD15 turns negative; (**F**–**J**) strong positivity of neoplastic cells for CD20, CD79a, PAX5/BSAP, and BOB1 (original magnification: ×4 for (**A**) and ×40 for all the remaining; (**A**–**C**) haematoxylin and eosin; (**D**–**J**): immunoalkaline phosphatase technique, Gill's haematoxylin nuclear counterstaining).

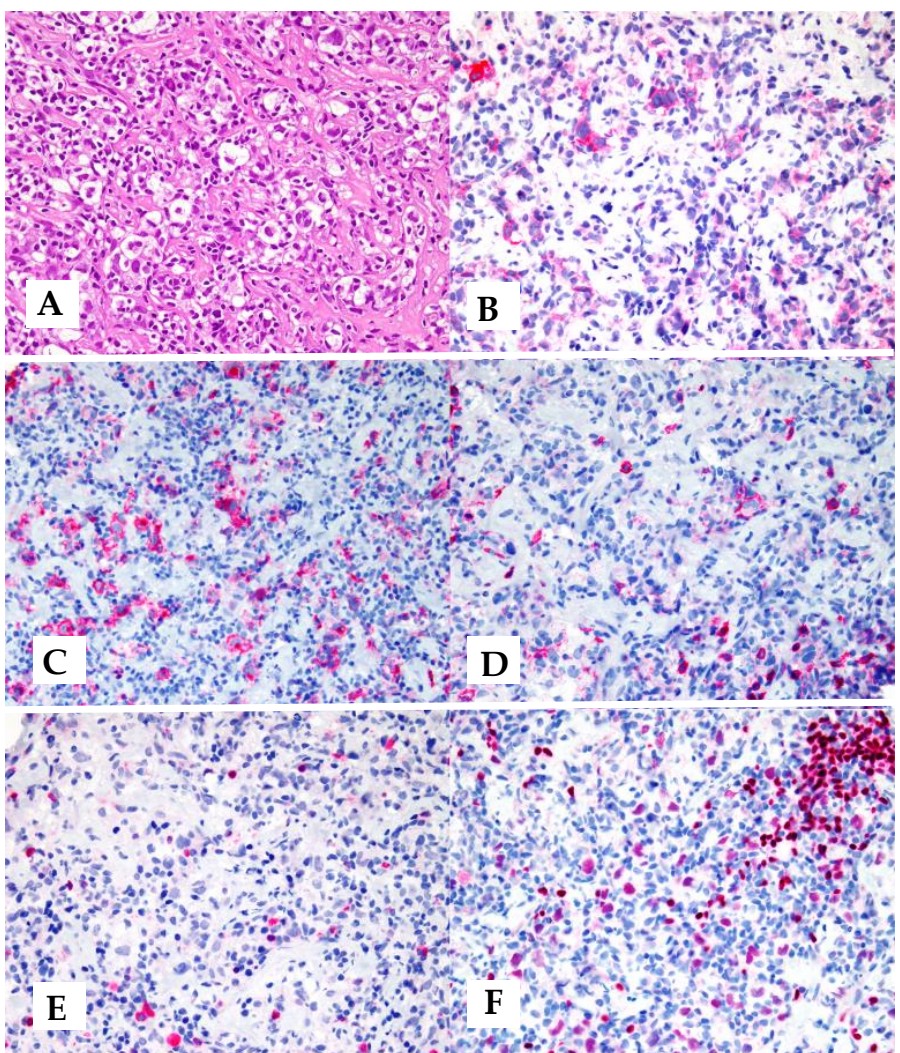

**Figure 2.** MGZL with PMBL-like morphology: (**A**) tumour consisting of large cells with clear cytoplasm, which are subdivided into clusters by a fibrotic reaction; (**B**,**C**) neoplastic cells express CD30 and CD15; (**D**,**E**) only a few tumoral cells turn positive for CD20, while CD79a staining is confined to scattered reactive plasma cells; (**F**) positivity for PAX5/BSAP varies from negative to moderate, always being lower than that of reactive small B-lymphocytes (original magnification: ×40; (**A**) haematoxylin and eosin; (**B**–**F**): immunoalkaline phosphatase technique, Gill's haematoxylin nuclear counterstaining).

## 6. Molecular Features

As in PMBL and CHL, alterations have been reported at the *JAK2/CD274/PDCD1LG2* locus at 9p24.1 and the *CIITA* locus at 16p13.13, including gains, amplifications, and rearrangements [19,20]. Alterations of the PD-L1 locus can represent the rational for the successful usage of immune checkpoint inhibitors, as discussed later. Gains at the *REL* locus at 2p.16.1 have been described in 33% of cases and gains at the *MYC* locus at 8q24 in 27% [21].

Methylation profiling of MGZL showed a distinct epigenetic profile intermediate between CHL and DLBCL but remarkably different from DLBCL [22]. By utilising genes such as *HOXA5*, *MMP9*, *EPHA7*, and *DAPK1* a final combined prediction of 100% was achieved among MGZL, CHL, and PMBL [22].

Sarkozy and coworkers carried out a gene expression profiling study in a large series of GZLs (mediastinal and extra-mediastinal), CHLs, PMBLs, and "polymorphic EBV+ DLBCLs of the NOS type" [8] (currently, EBV + DLBCL, NOS and EBV+ DLBCL in the ICC

and WHO-HEMA5, respectively) [10,11]. In an unsupervised principal component analysis, MGZLs showed intermediate scores in a spectrum between CHL and PMBCL, whereas EBV+ DLBCLs, NOS clustered distinctly. The main biological pathways underlying the MGZL spectrum were related to the cell cycle, which reflected the tumour cell content and extracellular matrix signatures that were related to TME [8]. Differential expression analysis and phenotypic characterisation of TME highlighted the predominance of regulatory macrophages in MGZL compared to CHL and PMBCL [8]. Notably, two distinct subtypes of GZL were distinguished and phenotypically reminiscent of PMBCL and DLBCL, respectively [8]. The former (PMBCL-type GZL) was characterised by clinical presentation in the "thymic" anatomic niche and actually corresponds to the present category of MGZL.

Sarkozy and coworkers also performed the first extensive next-generation sequencing (NGS) study of GZL and related entities [9]. In particular, they studied coding sequence mutations of 50 EBV-negative GZLs (including mediastinal "thymic" and nonmediastinal "extra-thymic" cases) and 20 EBV+ DLBCLs, NOS and compared them to examples of CHL, PMBL, and DLBCL [9]. Exomes of 21 GZL and 7 EBV+ DLBCL, NOS cases, along with paired constitutional DNA, were analysed in a discovery cohort, followed by targeted sequencing of 217 genes in an extension cohort of 29 GZLs and 13 EBV+ DLBCLs, NOS. MGZL cases with thymic niche involvement exhibited a mutation profile closely resembling CHL and PMBCL, *SOCS1* (45%), *B2M* (45%), *TNFAIP3* (35%), *GNA13* (35%), *LRRN3* (32%), and *NFKBIA* (29%) being the most recurrently mutated genes [9]. Such an assorted mutational profile of MGZL might correspond to three of the genetic subgroups (EZB, A53, and B2N) in the probabilistic classification of DLBCLs proposed by Wright et al. [23]. In contrast, cases formerly included within the GZL category but without thymic niche involvement (n = 18) had a significantly distinct pattern that was enriched in mutations related to apoptosis defects (*TP53* (39%), *BCL2* (28%), and *BIRC6* (22%)) and relatively depleted of mutations in *GNA13, XPO1*, or NF-kB signalling pathway genes (*TNFAIP3, NFKBIE, IKBKB*, and *NFKBIA*) [9]. They also exhibited more *BCL2/BCL6* rearrangements compared with thymic GZL. EBV+ DLBCLs, NOS presented a distinct mutational profile, including *STAT3* mutations and a significantly lower coding mutation load in comparison with EBV- MGZLs [9].

The two abovementioned contributions of the French Group, nicely commented by Elias Campo and Elaine S. Jaffe in *Blood*, have guided the decision to limit the concept of MGZL to EBV-negative mediastinal tumours in the ICC [10,12].

Since clonal intratumour heterogeneity may represent a driving force in MGZL plasticity and drug resistance, our group recently performed high-coverage targeted next-generation sequencing of eight primary and three refractory/relapsed MGZLs to better understand MGZL molecular complexity (manuscript in preparation). We subsequently identified genes under positive selection and inferred their clonal/subclonal structure through computational analysis. Genomic alterations occurring in low cancer cell fractions represented the majority of all events, suggesting that MGZL harbours a highly subclonal structure in both pre- and post-therapy scenarios. Evolutionary selection analysis identified several known genes involved in CHL and PMBL pathogenesis as putative drivers under positive selection (among them *SOCS1, B2M, TNFAIP3*, and *ITPKB*), whereas "neutral" genes were largely represented by epigenetic modifiers (such as *ARID1A, KMT2D, CREBBP, EP300, HIST1H1E*, and *HIST1H1C*). The observed findings suggest that MGZL harbours a complex clonal structure. Mutations affecting epigenetic controllers appear to be neutral events that have likely arisen early in tumour evolution. Conversely, mutations affecting genes under positive selection would represent subclonal events at diagnosis that subsequently expand their estimated clone size after therapy, possibly underpinning mechanisms of drug resistance (Figure 3).

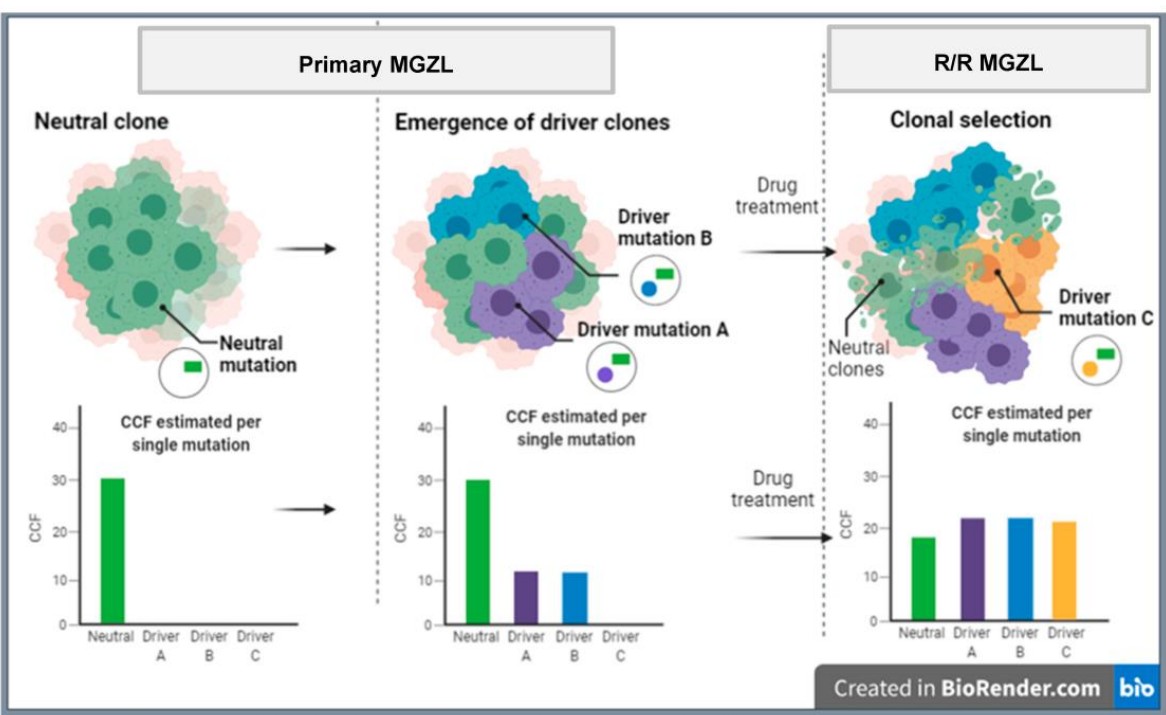

**Figure 3.** Hypothetical representation of clonal evolution in MGZL. R/R: relapsed/refractory; CCF: fraction of cancer cells harbouring each genetic event.

Indeed, further studies are required to unravel the molecular characteristics of MGZL in light of its poor response to current therapies.

## 7. Therapeutic Strategies

There is no standard therapy for MGZL [15,24]. Among 112 patients treated across 19 North American centres over a ten-year period, two regimens were predominately used: (A) cyclophosphamide, doxorubicin, oncovin, and prednisone (CHOP) +/− rituximab; (B) doxorubicin, bleomycin, vinblastine, and dacarbazine (ABVD) +/− rituximab. The overall response (ORR) and complete response (CR) rates were 82% and 73% for the patients who received rituximab, respectively, vs. 59% and 43% for those who did not, respectively [25]. Notably, the progression-free survival (PFS) rate was significantly inferior for patients treated with ABVD +/− rituximab compared to patients treated with CHOP +/− rituximab [25]. Recently, DA-EPOCH-R has been proposed as front-line therapy based on the efficacy shown in PMBL [26]. The results are better than the ones recorded with R-CHOP but not as excellent as in PMBL with a 5-year event-free survival (EFS) of 62% (PMBL 93%) and an overall survival (OS) of 74% (PMBL 97%) [26]. Thus, MGZL shows, overall, a worse prognosis than PMBL and CHL [15,24–26]. The role of radiotherapy is debated [15]. It has at times been used as a consolidative tool after chemoimmunotherapy for localised and/or bulky disease. However, no conclusion can be drawn because of the relatively small case number. The majority of patients with R/R disease have had hematopoietic stem cell transplantation (HSCT), more often autologous (ASCT). The 2-year OS of these patients was 88% vs. 67% for those who did not receive HSCT. However, caution should be used in analysing such results, since fit patients with chemosensitive disease were likely more frequently brought to HSCT [15,21–24]. Based on the results observed in PMBL [27], a very recent publication has reported that the combination of brentuximab vedotin (BV), and nivolumab is effective in approximately 70% of patients with R/R MGZL [28]. BV is an anti-CD30 humanised monoclonal antibody conjugated with monomethyl auristatin E, and the expression of CD30 by tumour cells represents the rationale of its usage [28,29]. BV was found to be effective also as maintenance after ASCT

in R/R MGZL [29]. In addition to the target effect, BV may also induce an immunogenic environment contributing to the depletion of regulatory T-cells that can potentiate the effect of immune-checkpoint inhibitors, such as nivolumab [28]. Nivolumab is an anti-PD1 monoclonal antibody that acts by restoring T-cell-mediated antitumour responses via blockade of PD-1/ligand interactions, a mechanism thought to be critical in MGZL, as supported by the findings of frequent copy number alterations of 9p24.1 and expression of PD-L1 and—less frequently—PD-L2 (see above) [22,27,28].

## 8. Conclusions and Future Directions

From what has been discussed above, MGZL, indeed, still represents a condition, to which the title of Luigi Pirandello's play "Six Characters in Search of an Author" can well apply. The rarity of the disease, the diagnostic difficulties, and the suboptimal response to most therapies underline the need for internationally shared guidelines. The complexity is further increased by the possibility that MGZL might not represent a single entity but rather a spectrum of diseases that require an individual tailoring of therapies. The 5th edition of the WHO Classification and ICC incorporate in their criteria an increased understanding of MGZL. Nonetheless, current studies are insufficient, and more in-depth molecular characterisation is needed to further understand the pathobiology of MGZL. In this respect, a significant contribution can be expected by the application of the new technologies and platforms allowing the molecular characterisation at the single cell level. It is likely that an improved understanding of genetic aberrations, microenvironmental characteristics, and cell-to-cell interactions in MGZL will lead to more effective targeted therapeutic approaches.

**Author Contributions:** Conceptualization, S.P.; review of the literature, V.T.; resources, V.T. and E.S.; data curation, A.C., F.M. and G.M.; writing—original draft preparation, S.P.; writing—review and editing, R.C., E.S. and P.L.Z.; visualization, S.P.; supervision, R.C., E.S., P.L.Z., M.R.S. and E.D.; project administration, S.P.; funding acquisition, S.P. All authors have read and agreed to the published version of the manuscript.

**Funding:** The manuscript was supported by AIRC 5x1000, grant number: 21198 (AIRC: Italian Association for Cancer Research, Milan, Italy).

**Institutional Review Board Statement:** The "unpublished data" were obtained in the course of two studies conducted according to the guidelines of the Declaration of Helsinki and approved by the Institutional Review Board of the European Institute of Oncology (protocol code: UID 2701) and the Ethical Committee of the Region Tuscany (CEAVC Em. 2022-263 Studio 18236_oss). The remaining parts of the manuscript represent a review of the literature, which required no approval.

**Informed Consent Statement:** Informed consent was obtained from all subjects involved in the studies, whose results are quoted as "unpublished data".

**Data Availability Statement:** The manuscript is a review of the literature. More details concerning the unpublished data can be asked to the corresponding author.

**Conflicts of Interest:** The authors declare no conflict of interest.

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
