# Peer review of "Mediastinal Gray-Zone Lymphoma: Still an Open Issue"

_hemato, doi:10.3390/hemato4030016_

Round 1
Reviewer 1 Report
The authors wrote the review literature on Gray zone lymphoma, one of the topics in the current lymphoma issues. The manuscript is well organized, however, the following points should be addressed.
1. In the following paragraph, other recent papers should be also discussed.
Clinical Characteristics 84 The median age of patients with GZL is 33 years and approximately two thirds are 85 males. Of these patients, nearly half have a bulky mediastinal mass >10 cm, while a mi- 86 nority have extranodal involvement and only 13% stage IV disease. The mediastinal mass 87 can cause superior vena cava syndrome or respiratory distress. Supra-clavicular lymph 88 nodes may be involved. There may be spread to lung by direct extension, as well as to 89 liver, spleen, and bone-marrow, in particular in relapsing or resistant (R/R) cases [14].
1) Blood Adv. 2017 Dec 11;1(26):2600-2609. 2017 Dec 12. Clinicopathologic consensus study of gray zone lymphoma with features intermediate between DLBCL and classical HL. Monika Pilichowska 1, Stefania Pittaluga 2, Judith A Ferry 3, Jessica Hemminger 4, Hong Chang 5 6, Jennifer A Kanakry 5, Laurie H Sehn 7, Tatyana Feldman 8, Jeremy S Abramson 9, Athena Kritharis 9, Francisco J Hernandez-Ilizaliturri 10, Izidore S Lossos 11, Oliver W Press 12, Timothy S Fenske 13, Jonathan W Friedberg 14, Julie M Vose 15, Kristie A Blum 16, Deepa Jagadeesh 17, Bruce Woda 18, Gaurav K Gupta 1, Randy D Gascoyne 7, Elaine S Jaffe 2, Andrew M Evens 19
2) Blood Adv. 2020 Jun 9; 4(11): 2523–2535. Gene expression profiling of gray zone lymphoma. Clémentine Sarkozy,1,2 Lauren Chong,2 Katsuyoshi Takata,2 Elizabeth A. Chavez,2 Tomoko Miyata-Takata,2 Gerben Duns,2 Adèle Telenius,2 Merrill Boyle,2 Graham W. Slack,2 Camille Laurent,3 Pedro Farinha,2 Thierry J. Molina,4 Christiane Copie-Bergman,5 Diane Damotte,6 Gilles A. Salles,1,7 Anja Mottok,8 Kerry J. Savage,2 David W. Scott,2 Alexandra Traverse-Glehen,1,9,* and Christian Steidlcorresponding author2,*
2. In figure 1, the second D) should be changed to E) ??
Author Response
Thank you very much indeed for your kind and constructive review.
1. The two suggested references have been added and discussed within the paragraph on clinical characteristics.
2. The mistake in the legend of Figure 1 has been amended.
The text changes have been evidenced in red on a yellow background in the revised manuscript.
Reviewer 2 Report
Pileri et al. summarized the recent progress of gray-zone lymphoma, based on the definition of ICC and WHO-5 classification. This review is well written, but several points are warranted to be addressed further.
1. The article by Traverse-Glehen, PMID 16224207, could be referred to in addition to the ref.13 as this is a representative study on the plasticity of GZL.
2. Line 79 Please elaborate on the results of GEP of mediastinal and non-mediastinal GZL, probably in the phenotype part or the molecular part. In the mediastinal GZL, PD-L1 and PD-L2 expression is higher and MHC is lower compared with non-thymic GZL.
3. Line 85. Authors mentioned that the median age of patients with GZL is 33 years and two thirds are males. Please give a reference for this data, distinguishing mediastinal and non-mediastinal GZL, and EBV-associated GZL, if possible.
4. Line 109. Could you give the information of CD15 in the Group 0?
5. Line 200-202. DLBCL has been subclassified into several genomic subtypes recently. Which subtype corresponds to the genomic profile of non-mediastinal GZL? EZB, A53, or B2N seem to suit.
6. Morphology and phenotype part: Several exceptional (pitfall) situations in the diagnosis of GZL are warranted to be introduced. First, syncytial variant of nodular sclerosis shows DLBCL-like morphology, but this is a distinct subtype of CHL. The phenomenon of “composite” and “sequential” lymphoma suggests the plasticity of PMBL/CHL, but this should not be included in GZL.
7. Figure 1 legend: line 3. Not “reach”, but rich. Line 4, not D) but E) for CD15.
8. Ref.14 and 21 are redundant.
Author Response
Thank you very much indeed for your careful and constructive review.
- The article of Traverse-Glehen has been quoted (new reference 14).
- The different expression of PDL-1/PD-L2 and MHC1 in mediastinal and non-mediastinal tumours has been added in the morphology and phenotypic section.
- The Section on clinical characteristics has been expanded as also suggested by Referee 1. Now three different references are quoted and commented.
- Information on CD15 expression in Group 0 has been added, based on what observed in the FIL study. In fact, this finding is not clearly stated in Sarkozy’s articles.
- The tentative correlation between the mutational landscape of MGZL and the genetic subtypes of DLBCL reported by Wright and coworkers has been added.
- The problem of the differential diagnosis between MGZL and “syncytial CHL” has been discussed in the section on morphology and phenotype.
- The legend of Figure 1 has been amended (rich instead of reach and E instead of D).
- In spite redundant, we decided to leave references 14 and 21, aiming to provide the reader as many pieces of information as possible.
The text changes have been evidenced in red on a yellow background in the revised manuscript.
Reviewer 3 Report
It was a pleasure to review the paper grey zone lymphoma. Congratulation on comprehensive review
Author Response
In the name of all authors, I wish to thank you very much indeed for your kind comments! Best wishes. Stefano Pileri
Reviewer 4 Report
This is a well-written review on the evolution of the concept of mediastinal grey zone lymphoma. The authors start at the very beginning when the discussion was largely on the practical differential diagnostic problem of overlapping features between CHL and DLBCL before moving to the convergence of the biological basis of the problem with overlap between CHL and PMBCL. From that time on, the terminology of MGZL was used by many quite commonly in daily practice despite not being formally introduced in the revised 4th WHO edition.
- Being so commonly used, it is confusing to see the abbreviation GZL being used throughout the manuscript. “Grey zone lymphoma has been applied in many more or less appropriate contexts and therefore it is not always obvious which GZL is addressed, also in this manuscript. This reviewer suggests to use the common terminology of MGZL throughout the manuscript to avoid confusion (even though admittedly historically not entirely correct and this reviewer apologizes to the first author, but still strong advises the use of MGZL).
- Morphology and immunophenotype: CHL-MGZL-PMBCL form a continuous biological, immunophenotypical and morphological spectrum. Although this fact is suggested at various instances, this should be more clearly phrased and explained. Importantly, the consequence that any separation of the three entities is thereby per definition arbitrary should be mentioned, especially in the context of (excellent) review of the various Sarkozy papers. Also, the conclusions that “more in depth molecular characterization is needed… “ (page 7, line 258-260) is meaningless in this respect. Please indicate what research is exactly needed and how and where this will contribute.
- A discussion (and opinion?) on EBV in the diagnosis of MGZL would be interesting. Does EBER+ excluded the diagnosis of MGZL in extramediastinal sites? And in mediastinal disease?
- Clinical characteristics: data seem to be based on one single review, please refer to primary papers and indicate ranges of age and stage distribution in various studies (and use “MGZL” instead of the confusing “GZL”).
- The statement of “assess whether there is any predictive test….” Brings up a subject not related to the rest of the paragraph and seems an alien object. Might be omitted.
- The biopsy issue: this is an important subject of discussion. However, the issue is not discussed in this section. Please discuss the differential diagnostic problem and the sampling problems (e.g. missing varying components, suboptimal sampling with sclerosis). Indeed, FNA is not suited for the precise classification of CHL/MGZL/PMBCL/DLBCL. It is very useful to exclude granulomatous disorders (sarcoidosis, TBC), metastatic disease, T-lymphoblastic lymphoma (in young males) etc. This issue may be added.
- Molecular features: please define what is meant with “polymorphic EBV+ DLBCL of the NOS-type”, since this is not an entity in any currently applied classification and not use what is meant. Please explain and use a more common terminology
- This reviewer sees it as a missed opportunity not to report on the NGS MGZL study performed by the Italian group (page6, line 209-216). The current text bears no relevant information. Either show the data or omit this section if data are too preliminary.
- Therapeutic strategies: The conclusion that MGZL appears to be relatively chemo-resistant….” does not follow from the information provided in the paragraph. Please explain and rephrase.
Minor:
correct English semantics prescribe classic HL, not classical (page 1, line 29)
High cell plasticity. High is not the correct wors, pleas rephrase (e.g. biological plasticity, phenotypic plasticity)
some minor editing may be needed
Author Response
Thank you very much indeed for your punctual and constructive comments.
- The term “MGZL” has been used, whenever it could be appropriately applied.
- The subjectivity of tracing the borders among PMBL, NSHL and GZL has been emphasized as well as the one among the groups proposed by Sarkozy and coworkers. At the end of the manuscript, a comment has been added concerning how the new single-cell technologies can improve the molecular knowledge of MGZL. On this respect, we have recently acquired the CosMx platform, which we intend to apply to several aggressive B-cell lymphomas, including GZLs, the aim being to better understand the microenvironmental composition as well as the cell-to-cell interactions.
- The EBV issue has been discussed by quoting what specifically stated in the International Consensus Classification.
- As suggested by Referee 1, we have expanded the clinical section by adding and commenting two additional papers.
- We have better explained why a parallel between THRLBCL and MGZL was made and how the approach applied to the former might represent a way to develop predictive models based on the microenvironmental characteristics in the setting of MGZL.
- The section dealing with the biopsy issue was expanded in the light of your useful comments.
- The term polymorphic EBV+ DLBCL of the NOS type introduced by Sarkozy and coworkers is now quoted in inverted commas and followed by the present terminology used by WHO-HEMA5 and ICC, the latter being used in the following.
- Concerning the results of our NGS analysis, we have expanded a bit the information in our hand and added a figure. Since the manuscript is in preparation, our findings would be fired by going into more details. We think that this highlight can be useful for the reader of the review to understand that the molecular issue cannot be regarded as definitively solved by the few studies so far published in the literature.
- The comment on chemoresistance has been deleted.
Minor:
- The adjective “classic” (referred to HL) is now used throughout the text.
- The expression “cell plasticity” has been substituted by “biological plasticity”.
- English style has been revised by a native, who is professor of English literature at Bologna University.
The text changes have been evidenced in red on a yellow background in the revised manuscript.